# Towards Stabilizable Sequential Smoothing Spline Interpolation by Point Forecasting

## Abstract

Sequential smoothing spline interpolators exhibit unstable behavior under low-delay response requirements. That is, instability issues are observed when a smoothing spline interpolator is forced to provide an interpolated trajectory piece subject to processing only a few to no incoming data points at each time stamp. Typically, the above instability setback is solved by increasing the delay, sacrificing some degree of smoothness in the interpolated trajectory, or a combination of both. However, stable sequential smoothing spline interpolation strategies working under low delay and without compromising their degree of smoothness seem vastly unexplored in the literature. To the best of our knowledge, this work formalizes the internal instability and asserts the controllability of sequential smoothing spline interpolators for the first time. Specifically, we model the trajectory assembled by a smoothing spline interpolator as a discrete dynamical system of the spline coefficients, facilitating the analysis of its internal instability and controllability. From these results, we propose a stabilizing strategy based on data point forecasting capable of operating even under delayless regimes and without sacrificing any smoothness of the interpolated trajectory. Our claims are theoretically confirmed, or experimentally supported by extensive numerical results otherwise.

## 1 Introduction

By interpolation, one typically refers to any estimation method aiming at computing an unobserved value, usually a data point, within the range of a set of observed data points. Among the plethora of estimators meeting the above definition, spline interpolators (Schoenberg, 1973; Späth, 1995) stand out in terms of popularity. This is arguably thanks to their model simplicity and outstanding approximation capacity to arbitrarily complex patterns. Successful applications comprise trajectory planning (Tordesillas & How, 2021; Marcucci et al., 2023), computer-aided design (Versprille, 1975), data compression (Lakshminarasimhan et al., 2011; Chiarot & Silvestri, 2023), and missing data imputation (Junger & De Leon, 2015; Biloš et al., 2022), among others.

Spline models are generally regarded as piecewise polynomial functions whose derivatives are continuous up to a certain derivative order. Despite this widely adopted conception, spline models can be found under (equivalent) different representations. For example, they can be expressed as a

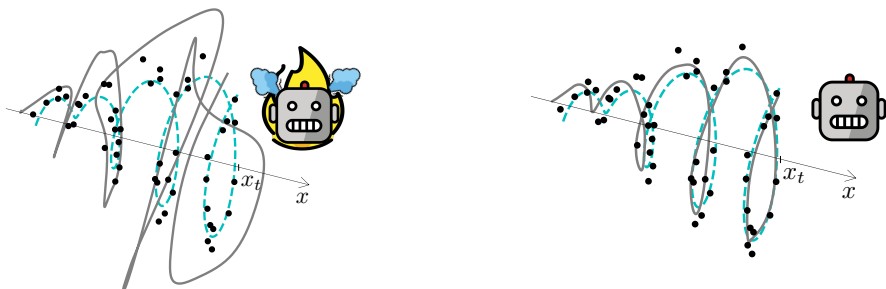

Figure 1: Visual comparison between an unstable sequential smoothing spline interpolator (left) and a stable one (right), over the same non-uniformly sampled and noisy stream of data points.

reproducing kernel Hilbert space (RKHS) basis expansion (Wahba, 1990; Nosedal-Sanchez et al., 2012) or as a minimal support basis expansion known as B-splines (Unser et al., 1993a;b), to name a few. More specifically, spline interpolators are usually determined as minimizers of an adequate measure of roughness (Wahba, 1978), e.g., the energy of the spline model, subject to interpolation constraints, i.e., to pass through all observed data points. Alternatively, smoothing spline interpolators are typically obtained as minimizers of a trade-off between the sum of squared residuals and the roughness measure (Eilers & Marx, 2010). From this perspective, spline interpolators can be viewed as a particular limit case of smoothing spline interpolators. Thus, the latter is often preferable, especially for noisy data.

Regardless of the chosen representation, the optimal computation of the smoothing spline interpolator requires the whole set of data points to be interpolated. This global character of smoothing spline interpolators can be problematic. Under some circumstances, e.g., while operating under a certain delay for streaming data, limited processing memory, or restricted computational capacity, accessing the whole set of observed data points becomes infeasible, or even impossible. In such cases, the corresponding smoothing spline interpolator cannot be determined in a straightaway manner. In fact, instability issues may arise if one tries to naively interpolate the streaming data using smoothing splines on the fly. That is, the sequentially interpolated trajectory may grow unbounded even for bounded streaming data, as shown in Figure 1.

Existing sequential smoothing spline interpolators for streaming data, trade-off some accuracy of the solution, i.e., of the interpolated trajectory, for the ability to operate in an online manner. In the literature, this is done by interpolating a serialized subset of adjacent data points using a windowed or delay mechanism, e.g., (de Carvalho & Hanson, 1986; Yu et al., 2010; Ogniewski, 2019). However, this delay mechanism poses an important drawback that must be taken into account for some spline representations in low-delay regimes. Taking the RKHS representation as an example, the overlapping between the (contributing) basis elements necessarily distorts the so-far interpolated trajectory when adding a new term to the basis expansion. Even B-splines having minimal support basis representation, suffer from this overlapping issue (Ruiz-Moreno et al., 2023b). Thus, a minimal delay is necessary and not all applications can afford it.

On the contrary, the piecewise spline representation is not affected by this impediment, since the spline pieces do not overlap between themselves by design. Nonetheless, this representation choice leads to an unstable behavior if one aims at sequentially minimizing the optimization objective that determines the interpolator while preserving the degree of smoothness in the overall interpolation. In this case, some strategies gain robustness or even overcome the instability issue by relaxing the optimization problem by just assuring a lower degree of smoothness. For instance, by using cubic Hermite splines (De Boor & De Boor, 1978), or variants (Debski, 2020; 2021; Ruiz-Moreno et al., 2023a). But again, some applications require a higher degree of smoothness, rendering these strategies insufficient.

Finally, there seems to be a lack of literature explicitly discussing the presented instability issue, either theoretically or practically.

**Contribution**. To the best of our knowledge, this paper formally describes for the first time the source of instability of sequential smoothing spline interpolators. Based on this, it elaborates a sequential smoothing spline interpolation strategy able to achieve stability, even under demanding low-delay regimes. Specifically, we first (equivalently) characterize the solution of the smoothing spline interpolation problem as the trajectory given by a discrete dynamical system of the spline coefficients. Then, we detect when such a dynamical system is internally unstable. This occurs for all smoothing spline configurations except for the linear case. Lastly, we corroborate its controllability and therefore our ability to stabilize the system.

The proposed strategy takes advantage of the stability of the (well-established) delay mechanism through data point forecasting. In this way, the predicted data points act as a surrogate of the data points that would have been observed after waiting for the corresponding delay. On the other hand, the local adaptability of spline models tends to favor equally: nearsighted forecasting methods at the data trend level and farsighted forecasting techniques at the data fluctuation level. Because of this, the former can be used without a significant drop in performance. Our strategy is successfully tested for several time series forecasting models and sources of streaming data points, thus experimentally validating its effectiveness.

**Organization**. The rest of the paper is organized as follows. Section 2 formulates the smoothing spline interpolation problem. Section 3 characterizes the dynamics of the sequential smoothing

spline interpolator, confirms its internal instability, and asserts its controllability. Then, the proposed stabilization strategy is described and validated in Section 4, and Section 5 concludes the paper. Finally, some complementary content can be found in the Appendices.

## 2 SMOOTHING SPLINE INTERPOLATION

In this Section, we introduce the standard formulation of the smoothing spline interpolation problem. Next, we present an alternative but equivalent formulation from a dynamic programming (DP) perspective (Bellman, 1966) which will allow us to better understand the dynamics of the solution to the sequential smoothing spline interpolation problem.

### 2.1 STANDARD FORMULATION

Given a space $\mathcal{W}_\rho$ of real functions defined over the domain $(x_0, x_{T+1}) \subseteq \mathbb{R}$ with $\rho - 1$ absolutely continuous derivatives and whose $\rho$th derivative is square integrable, and a set of data points $\{(x_t, y_t)\}_{t=1}^T$, with time stamps satisfying $x_0 < x_1 < \cdots < x_T < x_{T+1}$, the smoothing spline interpolation problem is usually formulated as

$$\min_{f \in \mathcal{W}_\rho} \left\{ \sum_{t=1}^T \ell\left(f(x_t), y_t\right) + \eta \int_{x_0}^{x_{T+1}} \left(D_x^\rho f(x)\right)^2 dx \right\}, \tag{1}$$

where $\ell : \mathbb{R} \to \mathbb{R}$ is a measure of fit that depends on $f$ only via $f(x_1), \ldots, f(x_T)$, and $D_x^\rho$ refers to the $\rho$th derivative with respect to (the variable) $x$.

The hyperparameters $\rho \in \mathbb{N}$ and $\eta \in \mathbb{R}_+$ are used to control the behavior of the solution to some extent. On the one hand, $\rho$ determines the search function space $\mathcal{W}_\rho$ and specifies the objective roughness, i.e., the second term in (1). On the other hand, $\eta$ controls the trade-off between the solution fit to the data points and its roughness. These hyperparameters can be used to accommodate prior knowledge about the solution, e.g., the minimum degree of smoothness of the solution. Preferably, they can be tuned to reduce the generalization error of the solution (Bischl et al., 2023). However, in this work, these hyperparameters are assumed given, thus leaving the question of how to tune them out of the scope of this paper. Accordingly, the hyperparameter dependencies in the notation across the paper are purposely omitted, when clear by context.

Provided that $T \geq \rho$, the unique solution to (1) is a univariate natural spline defined with the aid of $T$ knots (Schoenberg, 1964a;b; Wahba, 1990). That is, a real function $f$ with the following properties:

i) $f \in \Pi^{\rho-1}$ for $x \in (x_0, x_1] \bigcup [x_T, x_{T+1})$,

ii) $f \in \Pi^{2\rho-1}$ for $x \in [x_t, x_{t+1}]$ and $t = 1, \ldots, T-1$, and

iii) $f \in C^{2\rho-2}$ for $(x_0, x_{T+1})$,

where $\Pi^k$ is the class of polynomials of degree $k$ or less, and $C^k$ is the class of functions with $k$ continuous derivatives.

### 2.2 DYNAMIC PROGRAMMING APPROACH

The smoothing spline interpolation problem can be formulated from a state-action perspective (Ruiz-Moreno et al., 2023a). From this point of view, the form of the optimal solution described in Section 2.1, i.e., the natural spline, is explicitly taken into account; the corresponding interpolator plays the role of an agent, and the set of interpolated data points, i.e., $\mathcal{O}_{1:T} := \{\boldsymbol{o}_t\}_{t=1}^T$ with $\boldsymbol{o}_t := (x_t, y_t)$, is considered as a sequence of observations of the environment. Specifically, the coefficients of the piecewise spline representation $\{\boldsymbol{a}_t\}_{t=1}^T \subset \mathbb{R}^{2\rho}$ are viewed as a sequence of actions and every $t$th interpolation state (or simply state) $\boldsymbol{\sigma}_t$ encodes the necessary information to resume the interpolation task from the previous time step $t - 1$. More specifically, the state dynamics are governed by the following state update mechanism

$$\boldsymbol{\sigma}_{t+1} = f_\sigma(\boldsymbol{\sigma}_t, \boldsymbol{a}_t; \boldsymbol{o}_t) := (x_t, \boldsymbol{e}_t), \tag{2}$$

where $\boldsymbol{e}_t \in \mathbb{R}^{2\rho-1}$ contains a scaled limit of the first $2\rho - 1$ derivative values of the interpolated trajectory when approaching the time stamp $x_t$ from the left.

Formally, problem (1) is equivalent to

$$\min_{\boldsymbol{e}\in\mathbb{R}^{2\rho-1}} \left\{ J\left(\boldsymbol{\sigma}_1; \mathcal{O}_{1:T}\right) : \boldsymbol{\sigma}_1 = (x_0, \boldsymbol{e}) \right\}, \tag{3}$$

where

$$J\left(\boldsymbol{\sigma}_t; \mathcal{O}_{t:T}\right) = \min_{\{\boldsymbol{a}_i \in \mathcal{A}(\boldsymbol{\sigma}_i)\}_{i=t}^{T}} \left\{ \sum_{i=t}^{T} k(\boldsymbol{\sigma}_i, \boldsymbol{a}_i; \boldsymbol{o}_i) \right\} \tag{4a}$$

$$\text{subject to: } \boldsymbol{\sigma}_{i+1} = f_\sigma(\boldsymbol{\sigma}_i, \boldsymbol{a}_i; \boldsymbol{o}_i), \, \forall i \in \{t, \ldots, T-1\}, \tag{4b}$$

denotes the optimal cost from a given interpolation state and the corresponding sequence of (remaining) observations. $\mathcal{A}(\boldsymbol{\sigma}_i) := \{\boldsymbol{a} \in \mathbb{R}^{2\rho} : [\boldsymbol{a}]_{1:2\rho-1} = \boldsymbol{e}_{i-1}\}$ denotes the set of admissible actions from a given state, i.e., those spline coefficients that satisfy the degree of smoothness of the optimal solution when resuming the interpolation task. Finally, the mapping $k$ computes the instantaneous cost, i.e., the weighted sum between a given measure of fit and the roughness, incurred by an interpolation piece. See Appendix B, for more details.

In many applications, rather than finding the initial boundary conditions, as in (1) and (3), they are imposed, e.g., due to physical constraints like the starting position, velocity, and so on. In such cases, the smoothing spline interpolation problem simplifies to evaluating $J\left(\boldsymbol{\sigma}_1; \mathcal{O}_{1:T}\right)$ where $\boldsymbol{\sigma}_1$ and $\mathcal{O}_{1:T}$ are given. This can be done by finding the optimal sequence of spline coefficients $\{\boldsymbol{a}_1^*, \ldots, \boldsymbol{a}_T^*\}$ through the following DP algorithm:

Set

$$\boldsymbol{a}_1^* = \arg\min_{\boldsymbol{a}\in\mathcal{A}(\boldsymbol{\sigma}_1)} \left\{ k\left(\boldsymbol{\sigma}_1, \boldsymbol{a}; \boldsymbol{o}_1\right) + J\left(f_\sigma(\boldsymbol{\sigma}_1, \boldsymbol{a}; \boldsymbol{o}_1); \mathcal{O}_{2:T}\right) \right\}, \tag{5}$$

and

$$\boldsymbol{\sigma}_2^* = f_\sigma\left(\boldsymbol{\sigma}_1, \boldsymbol{a}_1^*; \boldsymbol{o}_1\right). \tag{6}$$

Then, sequentially going forward for $t = 2, \ldots, T$, set

$$\boldsymbol{a}_t^* = \arg\min_{\boldsymbol{a}\in\mathcal{A}(\boldsymbol{\sigma}_t^*)} \left\{ k\left(\boldsymbol{\sigma}_t^*, \boldsymbol{a}; \boldsymbol{o}_t\right) + J\left(f_\sigma(\boldsymbol{\sigma}_t^*, \boldsymbol{a}; \boldsymbol{o}_t); \mathcal{O}_{t+1:T}\right) \right\}, \tag{7}$$

and

$$\boldsymbol{\sigma}_{t+1}^* = f_\sigma\left(\boldsymbol{\sigma}_t^*, \boldsymbol{a}_t^*; \boldsymbol{o}_t\right). \tag{8}$$

## 3 DYNAMICS OF THE ACTION UPDATE MECHANISM

The DP algorithm presented in Section 2.2 can help us to model the dynamics of the sequence of actions. From here, we can study the internal stability as well as the controllability by borrowing well-established methodologies from control theory (Kirk, 2004; Doyle et al., 2013), as follows.

The DP step (7) can be rewritten as

$$\boldsymbol{a}_t^* = \boldsymbol{A}_{t-1}\boldsymbol{a}_{t-1}^* + \boldsymbol{B}\alpha_{t-1}^*, \tag{9}$$

where the input mapping

$$\alpha_t^* := \alpha\left(\boldsymbol{\sigma}_t^*; \mathcal{O}_{t:T}\right) \tag{10a}$$

$$= \arg\min_{\alpha\in\mathbb{R}^\rho} \left\{ k(\boldsymbol{\sigma}_t^*, \boldsymbol{C}\boldsymbol{e}_{t-1}^* + \boldsymbol{B}\alpha; \boldsymbol{o}_t) + J^*\left(f_\sigma(\boldsymbol{\sigma}_t^*, \boldsymbol{C}\boldsymbol{e}_{t-1}^* + \boldsymbol{B}\alpha; \boldsymbol{o}_t); \mathcal{O}_{t+1:T}\right) \right\}, \tag{10b}$$

results from redefining the optimization variable in the DP step (7) by incorporating the smoothness constraints, i.e., $[\boldsymbol{a}_t]_{1:2\rho-1} = \boldsymbol{e}_{t-1}^*$, into the optimization objective with the help of auxiliary matrices

$$\boldsymbol{C} := \left[\boldsymbol{I}_{2\rho-1}, \boldsymbol{0}_{(2\rho-1)\times 1}\right]^T \in \mathbb{R}^{2\rho\times(2\rho-1)}, \tag{11a}$$

$$\boldsymbol{B} := \left[\boldsymbol{0}_{1\times(2\rho-1)}, \boldsymbol{I}_1\right]^\top \in \mathbb{R}^{2\rho\times 1}, \tag{11b}$$

and where the matrix $\boldsymbol{A}_{t-1} \in \mathbb{R}^{2\rho\times 2\rho}$ captures the relation between the two consecutive actions $\boldsymbol{a}_{t-1}^*$ and $\boldsymbol{a}_t^*$, under the smoothness constraints. Explicitly, each element $[\boldsymbol{A}_t]_{i,j} =$

$u_t^{j-i-1} \prod_{\ell=1}^i (j - \ell)$ with $u_t := x_t - x_{t-1}$, except for the last row, in which $[\boldsymbol{A}_t]_{2\rho,j} = 0$. Visually,

$$
\boldsymbol{A}_t = \begin{bmatrix}
1 & u_t & u_t^2 & \cdots & u_t^{2\rho-1} \\
0 & 1 & 2u_t & \cdots & (2\rho - 1)u_t^{2\rho-2} \\
0 & 0 & 2 & \cdots & (2\rho - 2)(2\rho - 1)u_t^{2\rho-3} \\
\vdots & \vdots & \vdots & \ddots & \vdots \\
\vdots & \vdots & \vdots & \ddots & u_t \prod_{i=1}^{2\rho-1}(2\rho - i) \\
0 & 0 & 0 & \cdots & 0
\end{bmatrix}, \text{ and } \boldsymbol{B} = \begin{bmatrix} 0 \\ 0 \\ 0 \\ \vdots \\ 0 \\ 1 \end{bmatrix}. \tag{12}
$$

Particularly, with $\alpha_t^*$ as input mapping, we obtain the optimal sequence of actions. However, these dynamics can be studied for (any other) different sequence of inputs $\alpha_1, \alpha_2, \ldots, \alpha_{t-1}$, i.e.,

$$
\boldsymbol{a}_t = \boldsymbol{A}_{t-1}\boldsymbol{a}_{t-1} + \boldsymbol{B}\alpha_{t-1}. \tag{13}
$$

How to choose such a sequence of inputs under low-delay regimes is of paramount importance in this work, and is the central topic discussed in the ensuing sections.

### 3.1 INTERNAL STABILITY

We can assert the internal stability of the action update mechanism (13) by studying how the system evolves without any external input, i.e., $\boldsymbol{a}_t = \boldsymbol{A}_{t-1}\boldsymbol{a}_{t-1}$. Specifically, we are interested in knowing if for any initial bounded action $\boldsymbol{a}_1$, i.e., $\|\boldsymbol{a}_1\| < \infty$, we reach a bounded final $t$th action $\boldsymbol{a}_t$, regardless of the number of time steps[1] $t$.

**Theorem 1** (Internal Stability). *The action update mechanism* (13) *is internally stable for $\rho = 1$ and internally unstable for any $\rho > 1$ regardless of the arrangement of (strictly increasing) time stamps.*

*Proof.* Formally, the action update mechanism mentioned above is internally stable iff

$$
\lim_{t \to \infty} \|\boldsymbol{a}_t\| = \lim_{t \to \infty} \|\boldsymbol{A}_{t-1}\boldsymbol{A}_{t-2} \cdots \boldsymbol{A}_1 \boldsymbol{a}_1\| < \infty, \tag{14}
$$

for all bounded $\boldsymbol{a}_1$.

By construction, every $u_i := x_i - x_{i-1} > 0$. Therefore, every $\boldsymbol{A}_i$ is an upper triangular matrix with strictly positive entries above the main diagonal. Additionally, all matrices in $\{\boldsymbol{A}_i\}_{i=1}^{t-1}$ share the same set of eigenvalues (the diagonal entries), i.e., $\mathrm{eig}(\boldsymbol{A}_i) = \{j!\}_{j=1}^{2\rho-1} \cup \{0\}$ for $i \in \{1, \ldots, t-1\}$. Thanks to these properties, the product $\boldsymbol{A}_{t-1}\boldsymbol{A}_{t-2} \cdots \boldsymbol{A}_1$ is another upper triangular matrix with strictly positive entries above the main diagonal and with $\mathrm{eig}(\boldsymbol{A}_{t-1}\boldsymbol{A}_{t-2} \cdots \boldsymbol{A}_1) = \{(j!)^{t-1}\}_{j=1}^{2\rho-1} \cup \{0\}$. Thus, even when the sequence of $u_i$ values is given such that the upper diagonal entries of $\boldsymbol{A}_{t-1}\boldsymbol{A}_{t-2} \cdots \boldsymbol{A}_1$ tend to a finite value, if there is at least one eigenvalue greater than the unit, the limit in (14) diverges for some bounded $\boldsymbol{a}_1$. This occurs for any natural $\rho$ value except for $\rho = 1$, i.e., linear interpolation. In that case, notice that $\max \mathrm{eig}(\boldsymbol{A}_{t-1}\boldsymbol{A}_{t-1} \cdots \boldsymbol{A}_1) = 1$ regardless of the $u_1, \ldots, u_{t-1}$ values. $\square$

### 3.2 CONTROLLABILITY

By controllability, we refer to the ability of the action update mechanism (13) to reach, in a finite number of time steps $t$, any possible action configuration $\boldsymbol{a}_{t+1}$ (by using a sequence of inputs, i.e., $\alpha_1, \alpha_2, \ldots, \alpha_t$), regardless of the initial action $\boldsymbol{a}_1$. Then, if controllability holds; or in other words, if the action update mechanism (13) is controllable, we can ensure bounded sequences of actions (even for an infinite time horizon), through an adequate sequence of inputs.

---

[1]For the sake of completeness, it is also implicitly assumed that the sequence of observations is bounded in the sense that $\max\{|y_i|\}_{i=1}^t < \infty$ and $\max\{u_i\}_{i=1}^t < \infty$ for all $t$.

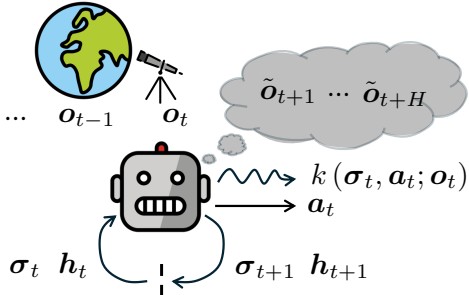

Figure 2: Diagram scheme of the proposed stabilization strategy for the sequential smoothing spline interpolation problem. The robot represents the interpolator, or agent. The telescope aiming at the globe symbolizes the sampling step. The thought bubble illustrates the data point forecasting procedure. Lastly, $\boldsymbol{h}_t$ stands for the agent memory (Zhang et al., 2016) at time step $t$, e.g., a hidden state (of fixed size) that encodes a representation of all previously observed data point values.

The controllability of the action update mechanism can be asserted by noticing that

$$\boldsymbol{a}_2 = \boldsymbol{A}_1 \boldsymbol{a}_1 + \boldsymbol{B}\alpha_1, \tag{15a}$$

$$\boldsymbol{a}_3 = \boldsymbol{A}_2\left(\boldsymbol{A}_1 \boldsymbol{a}_1 + \boldsymbol{B}\alpha_1\right) + \boldsymbol{B}\alpha_2, \tag{15b}$$

$$\vdots$$

$$\boldsymbol{a}_{t+1} - \boldsymbol{A}_t \boldsymbol{A}_{t-1} \cdots \boldsymbol{A}_1 \boldsymbol{a}_1 = \left[\boldsymbol{A}_t \boldsymbol{A}_{t-1} \cdots \boldsymbol{A}_2 \boldsymbol{B}, \ldots, \boldsymbol{A}_t \boldsymbol{B}, \boldsymbol{B}\right] \boldsymbol{\alpha}_t, \tag{15c}$$

where $\boldsymbol{\alpha}_t := \left[\alpha_1, \alpha_2 \ldots, \alpha_t\right]^\top \in \mathbb{R}^t$. Specifically, the left-hand side term in (15c) can always be set zero-valued iff the action update mechanism is controllable; or equivalently, if the matrix $\boldsymbol{M}_t := \left[\boldsymbol{A}_t \boldsymbol{A}_{t-1} \cdots \boldsymbol{A}_2 \boldsymbol{B}, \ldots, \boldsymbol{A}_t \boldsymbol{B}, \boldsymbol{B}\right] \in \mathbb{R}^{2\rho \times t}$ is full row rank, i.e., $\mathrm{rank}(\boldsymbol{M}_t) = 2\rho$.

**Result 1.** *The action update mechanism* (13) *is controllable for $\rho = 2$ regardless of the sequence of $u_t$ values.*

*Proof.* It suffices to verify that $\boldsymbol{M}_4 = \left[\boldsymbol{A}_4 \boldsymbol{A}_3 \boldsymbol{A}_2 \boldsymbol{B}, \boldsymbol{A}_4 \boldsymbol{A}_3 \boldsymbol{B}, \boldsymbol{A}_4 \boldsymbol{B}, \boldsymbol{B}\right] \in \mathbb{R}^{4 \times 4}$ is full rank. Since

$$\det(\boldsymbol{M}_4) = 864 u_2^2 u_3 u_4^3 - 216 u_2^2 u_4^4 - 432 u_2 u_3 u_4^4 + 90 u_2 u_4^5 + 216 u_3^2 u_4^4 - 36 u_3 u_4^5 \tag{16}$$

cannot be set to zero for any $u_2, u_3, u_4 > 0$, we can conclude that $\mathrm{rank}(\boldsymbol{M}_4) = 4$. $\qquad\square$

**Conjecture 1.** *The action update mechanism* (13) *is controllable for any $\rho$ regardless of the sequence of $u_t$ values.*

We numerically support **Conjecture 1** for each $\rho \in \{3, \ldots, 10\}$ for 1000 different realizations of $\{u_t \sim \mathcal{U}(10^{-3}, 2)\}_{t=2}^{2\rho}$.

From **Result 1**, we are guaranteed that the sequential smoothing spline interpolator is controllable, hence stabilizable for cubic splines, i.e., for $\rho = 2$. Cubic splines are most commonly used to exploit the local approximation capacity of splines (de Carvalho & Hanson, 1986). This is because, as the degree of the spline increases the spline interpolation will behave more like a polynomial fit. However, if a higher degree of smoothness is needed, e.g., an interpolated trajectory in $C^{2\rho-2}$, **Conjecture 1** extends the **Result 1** for any natural $\rho$ value.

## 4 A STABILIZATION STRATEGY

Section 3 provides a theoretical framework that can describe the sequential smoothing spline interpolation instability phenomena. It also outlines that a sequential smoothing spline interpolation procedure can be stabilized by adequately choosing the inputs that determine the spline coefficients. Moreover, thanks to the equivalence between the standard and the DP formulations presented in Section 2.1 and 2.2 respectively, the optimal input mapping (10) guarantees a stable sequence of actions as long as the result of (1) is bounded. Therefore, approximating such an optimal input mapping

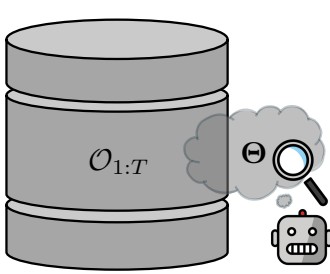 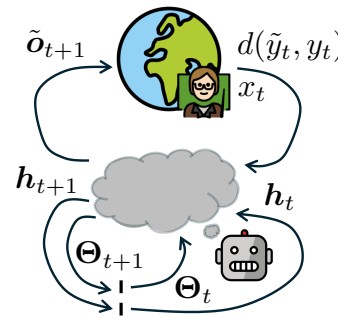

Figure 3: Diagram scheme of the offline learning setting (left) and the online learning setting (right) for the parametrized predictors discussed in Section 4.2.2. The teacher exemplifies the computation of any distance mapping $d$ (in this case, for $H = 1$) between the forecasted data point value $\tilde{y}_t$ and the actual one $y_t$.

seems a sensible starting point for developing stabilizing input strategies when optimality cannot be achieved.

A relevant study case in which the optimal input mapping cannot be readily evaluated is the family of smoothing spline interpolation problems subject to delay response requirements. This is because at any $t$th time step, future data points $\mathcal{O}_{t+1:T}$ may be unavailable; that is, the wait needed to receive them violates the delay constraints. Consequently, the optimal cost-to-go (4) cannot be evaluated, and by extension neither the optimal input mapping (10).

Notice that the dynamic programming step (7) is implicitly describing the following policy evaluation,

$$\pi\left(\boldsymbol{\sigma}_t, \mathcal{O}_{t:T}\right) := \arg\min_{\boldsymbol{a} \in \mathcal{A}(\boldsymbol{\sigma}_t)} \left\{k\left(\boldsymbol{\sigma}_t, \boldsymbol{a}; \boldsymbol{o}_t\right) + J\left(f_\sigma(\boldsymbol{\sigma}_t, \boldsymbol{a}; \boldsymbol{o}_t); \mathcal{O}_{t+1:T}\right)\right\}, \tag{17}$$

or equivalently, $\boldsymbol{a}_t = \pi\left(\boldsymbol{\sigma}_t, \mathcal{O}_{t:T}\right) = [\boldsymbol{e}_{t-1}^\top, \alpha(\boldsymbol{\sigma}_t; \mathcal{O}_{t:T})]^\top$. Next, we describe and justify an approximation strategy to (17) capable of achieving stability under low-delay requirements.

## 4.1 STABILIZATION THROUGH DELAY

Suppose the delay constraint allows for waiting for $L$ data points before forcing us to take action. In that case, a reasonable strategy may be following an $L$-step look-ahead policy (Bertsekas, 2019). In other words, a policy that computes the cost until the $L$th data point instead of the (whole) cost-to-go, i.e., $\pi\left(\boldsymbol{\sigma}_t, \mathcal{O}_{t:T+L}\right)$.

This strategy has been successfully used in the literature in several forms, e.g., the windowed mechanism presented in Section 1. Moreover, if for some delay the action update mechanism still presents instability, it can usually be solved by further increasing the delay.

## 4.2 STABILIZATION THROUGH PREDICTION

Now, let us suppose that the delay constraint does not allow for waiting for any additional data points; that is, $L = 0$. In that case, it has been observed that the myopic policy, i.e., $\pi(\boldsymbol{\sigma}_t; \boldsymbol{o}_t) = \arg\min\left\{k\left(\boldsymbol{\sigma}_t, \boldsymbol{a}; \boldsymbol{o}_t\right) : \boldsymbol{a} \in \mathcal{A}(\boldsymbol{\sigma}_t)\right\}$ suffers from severe instability (Ruiz-Moreno et al., 2023a).

Taking into account that the delay can help to stabilize the action update mechanism, one plausible strategy is to forecast future data points and use them as if they were part of a delayed sequence of data points to evaluate the policy, i.e., $\pi(\boldsymbol{\sigma}_t; \{\boldsymbol{o}_t\} \cup \tilde{\mathcal{O}}_{t+1:t+H})$ where $\tilde{\mathcal{O}}_{t+1:t+H}$ is the set of the predicted next $H$ data points. This is illustrated in Figure 2.

Since predicting the next $H$ data points can be formulated as a time-series forecasting problem, most of the available time-series prediction models in the literature can be directly borrowed. Naturally, the further and more accurate predictions one can have, the closer the policy evaluation gets to the optimal policy and therefore, the closer to guaranteed stability.

On the other hand, the trend of the data plays a decisive role and is enough information in practice.

That is, a prediction model able to capture relevant aspects of the trend performs relatively close to optimal. This argument is further developed and experimentally supported in Appendix C.

Based on this, the chosen prediction models are relatively rudimentary but practical. They are also representative of more sophisticated or problem-tailored approaches. That is, if a simple approach with limited prediction capacity can achieve stability a more accurate strategy will also achieve it. In this work, we consider a parameterless and a parameterized predictor. Within the parameterized one, we also discuss and study whether offline or online training can be carried out while preserving stability.

### 4.2.1 TIME STAMPS LOCATION FORECASTING

Recall that the observation time stamps $x_1, x_2, \ldots, x_T$ considered in this work are not necessarily uniformly arranged. Because of this, at time step $t$, we estimate the time stamps of the observations to come, i.e., $\tilde{x}_{t+1}, \ldots, \tilde{x}_{t+H}$, or more specifically their relative time distance to the previous observation time stamp, i.e., $\tilde{u}_{t+1}, \ldots, \tilde{u}_{t+H}$, as the moving average. Formally, we set $\tilde{u}_{t+1} = \cdots = \tilde{u}_{t+H} = \mu_t$, where

$$\mu_t = \frac{t-1}{t}\mu_{t-1} + \frac{1}{t}u_t, \tag{18}$$

which leads to $\tilde{x}_i = x_t + (i - t)\mu_t$ for $i = t + 1, \ldots, t + H$.

### 4.2.2 DATA POINT VALUES FORECASTING

The simplest approach to time-series forecasting is arguably the zero-order hold (ZOH) digital-to-analog conversion model (Pelgrom, 2013) evaluated at the predicted time stamps, i.e., $\tilde{y}_{t+1} = \cdots = \tilde{y}_{t+H} = y_t$.

Alternatively, we explore a linear model, as arguably the simplest representative of parametric models, of the form

$$\tilde{\boldsymbol{y}}_{t+1:t+H} = \boldsymbol{\Theta}\,\boldsymbol{y}_{t-P+1:t}, \tag{19}$$

where $\tilde{\boldsymbol{y}}_{t+1:t+H} := [\tilde{y}_{t+1}, \ldots, \tilde{y}_{t+H}]^\top \in \mathbb{R}^H$ denote the $H$-forecasted data point values, $\boldsymbol{y}_{t-P+1:t} := [y_{t-P+1}, \ldots, y_t]^\top \in \mathbb{R}^P$, refers to the $P$-lagged data point values considered for the prediction, and $\boldsymbol{\Theta} \in \mathbb{R}^{H \times P}$ contains the learnable parameters of the model.

We are also interested in the possible effect of the parameters training on stability. Therefore, we consider an offline learning setting in which the optimal parameters are obtained from a training dataset and an online learning setting in which the parameters are updated as data points are received, as illustrated in Figure 3. This can be done via least squares (LS) (Lawson & Hanson, 1995), or via recursive least squares (RLS) (Kailath et al., 2000).

The solution to the associated LS problem is obtained as

$$\boldsymbol{\Theta} = \widetilde{\boldsymbol{Y}}^\top \boldsymbol{Y} \left(\boldsymbol{Y}^\top \boldsymbol{Y}\right)^{-1}, \tag{20}$$

where $\widetilde{\boldsymbol{Y}}$ and $\boldsymbol{Y}$ denote the matrices whose rows are $\boldsymbol{y}_{t+1:t+H}^\top$ and $\boldsymbol{y}_{t-P+1:t}^\top$ for $t = P, \ldots, T - H$, respectively.

On the other hand, the RLS updates are computed as

$$\boldsymbol{\Theta}_t = \boldsymbol{\Theta}_{t-1} + \left(\tilde{\boldsymbol{y}}_{t+1:t+H} - \boldsymbol{\Theta}_{t-1}\boldsymbol{y}_{t-P+1:t}\right)\boldsymbol{y}_{t-P+1:t}^\top\boldsymbol{\Sigma}_t^{-1}, \tag{21a}$$

$$\boldsymbol{\Sigma}_t^{-1} = \boldsymbol{\Sigma}_{t-1}^{-1} - \boldsymbol{\Sigma}_{t-1}^{-1}\boldsymbol{y}_{t-P+1:t}\left(1 + \boldsymbol{y}_{t-P+1:t}^\top\boldsymbol{\Sigma}_{t-1}^{-1}\boldsymbol{y}_{t-P+1:t}\right)^{-1}\boldsymbol{y}_{t-P+1:t}^\top\boldsymbol{\Sigma}_{t-1}^{-1}, \tag{21b}$$

where $\boldsymbol{\Theta}_t \in \mathbb{R}^{H \times P}$ contains the estimated parameter values, and $\boldsymbol{\Sigma}_t \in \mathbb{R}^{P \times P}$ is the estimated sample covariance matrix, at time step $t$.

### 4.3 EXPERIMENTS

The set of observations $\mathcal{O}_{1:T}$ to be interpolated is generated as a realization of a (stable) autoregressive process (Mills, 1990) AR(2) over unitary uniformly arranged values, i.e., $u_t = 1$ for all $t \in \mathbb{N}^{[1,T]}$, with $T = 300$. Specifically, as $y_{t+1} = \boldsymbol{\phi}^\top[y_t, y_{t-1}]^\top + w_t$ with $y_0 = 0$, autoregressive

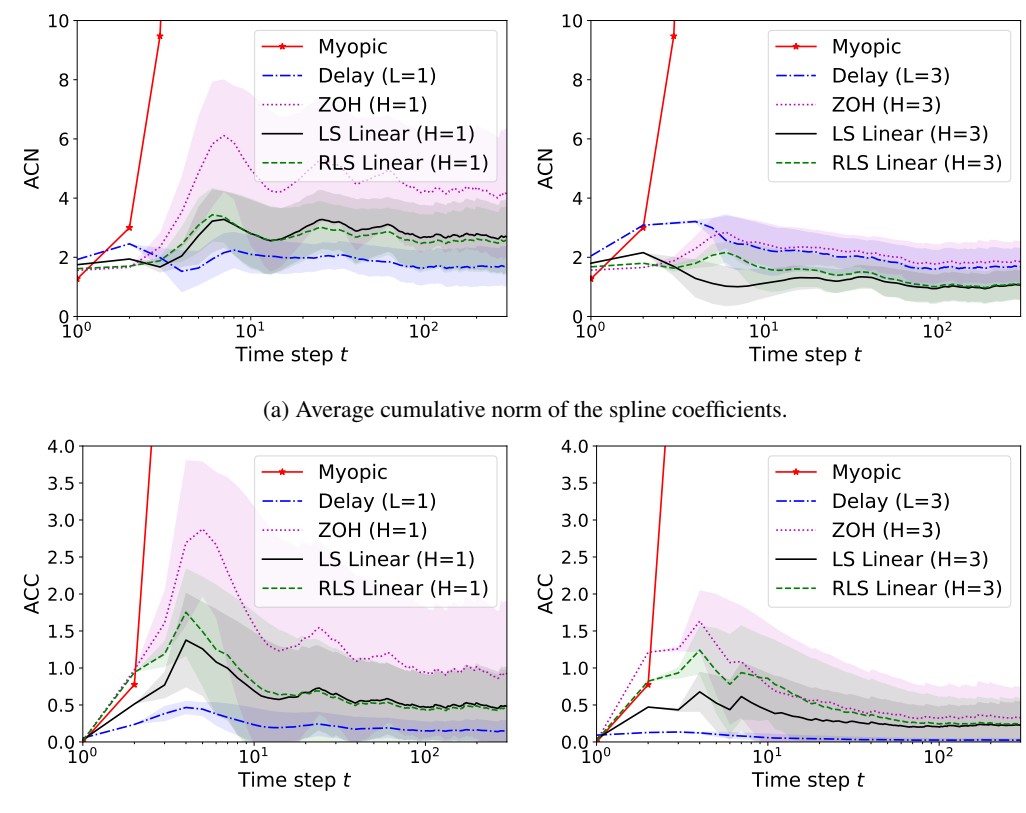

(a) Average cumulative norm of the spline coefficients.

(b) Average cumulative cost incurred by the interpolator.

Figure 4: The average and standard deviation (shaded area) over time of the cumulative norm (of the spline coefficients) and cumulative cost (incurred by the interpolator) remain bounded showing that the proposed interpolators are stable. On the contrary, the myopic strategy diverges at very early time steps for both metrics.

coefficients (randomly chosen) $\phi = [0.40235509, 0.52433128]^\top$ and $w_t \sim \mathcal{N}(0, 0.60276338)$. The same AR(2) process, but using a different realization, has been used to generate training data of $T = 500$ data points. Recall that this training data is only used for the LS linear forecasting strategy.

Particularly, we have used cubic splines, i.e., $\rho = 2$ and hyperparameter $\eta = 0.01$, for all strategies discussed in Section 4. The myopic and the prediction-based strategies work under no delay, i.e., $L = 0$. The ZOH strategy uses only the previous data point value, $P = 1$, while the linear predictors use the same lagged values as the original AR(2) process, i.e., $P = 2$.

In order to validate their effectiveness, we use the average cumulative norm (ACN), i.e., $\frac{1}{t} \sum_{i=1}^{t} \|a_i\|_2$, and average cumulative cost (ACC), i.e., $\frac{1}{t} \sum_{i=1}^{t} k(\sigma_i, a_i; o_i)$, as numerical stability proxies. This is because, for a stable interpolator, both metrics are bounded as $t$ increases.

As shown in Figure 4, all discussed strategies except the myopic one are numerically stable. As expected, the delay mechanism achieves the lowest ACC since it uses the exact data points to come. Moreover, we observe that increasing the prediction horizon $H$ can help to reduce the ACN of the spline coefficients (hence, dumping oscillations) while improving the ACC as well as reducing its variability (standard deviation). However, it can be observed that the prediction horizon $H$ tends to dim the incremental improvement over the ACC, as $H$ increases. In other words, the benefits of large $H$ values on reducing the ACC are marginal. This is also expected, due to the local approximation capacity of splines; that is, neighboring data points have a larger influence in the interpolation step than farther ones.

For additional complementary experiments over the same experimental setting, covering aspects such as the execution time or some insights into the significance of the accuracy and horizon length

of the predictor, we refer the reader to the Appendices C.2 and C.3. Similarly, for different experiment setting variations or data generated from an arguably more challenging data process see Appendices C.1 and C.4, respectively.

## 5 CONCLUSION

This work is meant as a first step towards stabilization strategies for the problem of sequential smoothing spline interpolation under few to no delay constraints. The proposed strategies for stable sequential smoothing spline interpolation are envisioned to be applied to streaming time-series data points sampled from smooth processes, under possibly no delay constraints. However, the sequential nature of the proposed strategies makes them equally useful for processing data sets that are too large to fit in memory and are accessed sequentially.

**Limitations**. Future work could be focused on proving formally **Conjecture 1** that affirms that any bounded trajectory regardless of its degree of smoothness (determined by $2\rho - 2$), can be controlled and thus, stabilized. Similarly, whether it is possible to delimit the effectiveness of the proposed stabilization strategies theoretically remains an open problem.

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

APPENDICES

## A  REMARKS

This section is meant to expand on some important concepts and implications indirectly covered in the main document.

### A.1  SAMPLE DELAY

The term delay is rather vague and admits several interpretations. For example, in the context of this work, it could have been used to express the time difference between observed data points at any time step, e.g., $x_t - x_{t-1}$, or the time needed by the interpolator to process the incoming data and produce its outcome, i.e., the time needed to interpolate a piece of the trajectory.

In this work, the term delay is exclusively used to refer to the number of (incoming or accessed) samples needed to be processed before a trajectory piece is proposed. In this sense, a sequential smoothing interpolator operating under no delay has to provide, at every time step $t$, the interpolated trajectory piece between any two consecutive data points $(x_{t-1}, y_{t-1})$ and $(x_t, y_t)$ without knowing any of the ensuing data points, i.e., $\boldsymbol{o}_{t+1}, \ldots, \boldsymbol{o}_T$.

### A.2  MULTIVARIATE TRAJECTORY INTERPOLATION

A (multivariate) trajectory can be expressed as $\boldsymbol{\psi}(x) = [\psi_1(x), \ldots, \psi_N(x)]^\top$, where every $\psi_n(x)$ describes a (univariate) curve along the $n$th dimension. In this work, we mainly refer to the index $x$ as time. However, it is worth noting that the trajectories we discuss here are not only limited to time functions.

When an $N$-variate trajectory $\boldsymbol{\psi}(x)$ is sampled at $x_1, \ldots, x_T$ we obtain $T$ data points, where every $t$th data point is of the form $\boldsymbol{o}_t := (x_t, \psi_1(x_t), \ldots, \psi_N(x_t))$. Notice that (if $N > 1$), we can split the sequence $\{\boldsymbol{o}_t\}_{t=1}^T$ into $N$ sequences of the form $\left\{\boldsymbol{o}_t^{(1)}\right\}_{t=1}^T, \ldots, \left\{\boldsymbol{o}_t^{(N)}\right\}_{t=1}^T$, where $\boldsymbol{o}_t^{(n)} := \left(x_t, y_t^{(n)}\right)$, and $y_t^{(n)} := \psi_n(x_t)$. In this way, each of the resulting $n$th sequences of data points describes the sampled curve in the corresponding $n$th dimension. Moreover, each of these new sequences can be interpolated separately, possibly in parallel. Then, the (multivariate) interpolated trajectory is obtained by combining the (univariate) interpolated curves.

The above preprocessing step allows us to employ the interpolation strategy presented in this paper to any multivariate trajectory. For example, this has been used for the target trajectory (dashed line) in Figure 1, which corresponds to $\boldsymbol{\psi}(x) = [\psi_1(x), \psi_2(x)]^\top$ where $\psi_1(x) = \frac{1}{2}log(x)sin(x/4)$, and $\psi_2(x) = \frac{1}{2}log(x)cos(x/4)$.

# B DYNAMIC PROGRAMMING ALGORITHM FOR THE SMOOTHING SPLINE INTERPOLATION PROBLEM

This section presents a step-by-step derivation of the dynamic programming algorithm for the smoothing spline interpolation problem presented in Section 2.2.

The solution to the smoothing spline interpolation problem (1) can be expressed in general as

$$
f(x) = \begin{cases} g_1(x), & \text{if } x \in (x_0, x_1] \\ \vdots \\ g_t(x), & \text{if } x \in (x_{t-1}, x_t] \\ \vdots \\ g_{T+1}(x), & \text{if } x \in (x_T, x_{T+1}) \end{cases} \tag{22}
$$

where every piece $g_t : (x_{t-1}, x_t] \to \mathbb{R}$ is a linear combination of polynomials of the form

$$
g_t(x) = \boldsymbol{a}_t^\top \boldsymbol{p}_t(x), \tag{23}
$$

with combination coefficients $\boldsymbol{a}_t \in \mathbb{R}^{2\rho}$ and basis vector function $\boldsymbol{p}_t : (x_{t-1}, x_t] \to \mathbb{R}^{2\rho}$ constructed as

$$
\boldsymbol{p}_t(x) = \left[ 1, x - x_{t-1}, (x - x_{t-1})^2, \dots, (x - x_{t-1})^{2\rho-1} \right]^\top. \tag{24}
$$

By using the form of the solution stated in (22), (23), and its properties stated in Section 2.1, the optimization problem (1) can be equivalently rewritten as

$$
\min_{\{\boldsymbol{a}_t \in \mathbb{R}^{2\rho}\}_{t=1}^T} \left\{ \sum_{t=1}^T \ell\left( \boldsymbol{a}_t^\top \boldsymbol{p}_t(x_t), y_t \right) + \eta \int_{x_{t-1}}^{x_t} \left( \boldsymbol{a}_t^\top \boldsymbol{p}_t^{(\rho)}(x) \right)^2 dx \right\} \tag{25a}
$$

$$
\text{subject to: } \lim_{x \to x_{t-1}^-} \boldsymbol{a}_{t-1}^\top \boldsymbol{p}_{t-1}^{(k)}(x) = \lim_{x \to x_{t-1}^+} \boldsymbol{a}_t^\top \boldsymbol{p}_t^{(k)}(x), \; \forall t = 2, \dots, T \text{ and } \forall k = 1, \dots, 2\rho - 2, \tag{25b}
$$

where superindexing the basis vector, i.e., $\boldsymbol{p}_t^{(k)}$, is shorthand notation for the $k$th derivative with respect to the variable $x$. Additionally, we have explicitly imposed our knowledge about the roughness of the last spline piece $g_{T+1}$, which is zero. This imposition is unnecessary since it is a property of the solution and therefore arises naturally. However, making it explicit will simplify algebraic manipulations in the next derivations.

Regarding the right-hand side of the equality constraints in (25b), it can be further simplified by using the following relation

$$
\left[ \boldsymbol{p}_t^{(k)}(x) \right]_i = (x - x_{t-1})^{i-1-k} \prod_{j=1}^k (i - j), \tag{26}
$$

because

$$
\lim_{x \to x_{t-1}^+} \left[ \boldsymbol{p}_t^{(k)}(x) \right]_i = \begin{cases} k! & \text{if } i = k + 1, \\ 0 & \text{otherwise.} \end{cases} \tag{27}
$$

In this way,

$$
\lim_{x \to x_{t-1}^+} \boldsymbol{a}_t^\top \boldsymbol{p}_t^{(k)}(x) = k! \left[ \boldsymbol{a}_t \right]_{k+1}, \tag{28}
$$

and

$$
\left[ \boldsymbol{a}_t \right]_i = \frac{1}{(i-1)!} \lim_{x \to x_{t-1}^-} \boldsymbol{a}_{t-1}^\top \boldsymbol{p}_{t-1}^{(i-1)}(x) \tag{29a}
$$

$$
\underset{(26)}{=} \frac{1}{(i-1)!} \sum_{j=1}^{2\rho} \left[ \boldsymbol{a}_{t-1} \right]_j (x_{t-1} - x_{t-2})^{j-1} \prod_{l=1}^{i-1} (j - l) \tag{29b}
$$

$$
:= \left[ \boldsymbol{e}_{t-1} \right]_i. \tag{29c}
$$

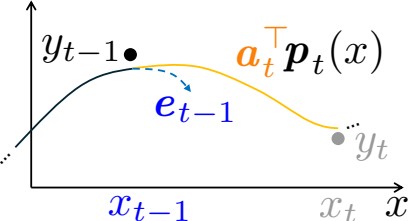

Figure 5: Action-state representation of the smoothing spline interpolation problem at time step $t$. The signal state components $x_{t-1}$ and $e_{t-1}$ are highlighted in blue, the $t$th spline coefficients in $a_t$ are illustrated in orange and the $t$th observation elements $x_t$ and $y_t$ are displayed in gray.

Equivalently,

$$[a_t]_{1:2\rho-1} = e_{t-1}, \tag{30}$$

where the components of every $e_{t-1} \in \mathbb{R}^{2\rho-1}$ are constructed as in (29b).

In a similar way, the second term in the cost (25a) can be further simplified by making use of the following relation

$$\int_{x_{t-1}}^{x_t} \left( a_t^\top p_t^{(\rho)}(x) \right)^2 dx = a_t^\top \left[ \int_{x_{t-1}}^{x_t} p_t^{(\rho)}(x) \, p_t^{(\rho)}(x)^\top dx \right] a_t \tag{31}$$

where we define $R_t \in S_t^{2\rho}$ as

$$R_t := \int_{x_{t-1}}^{x_t} p_t^{(\rho)}(x) \, p_t^{(\rho)}(x)^\top dx, \tag{32}$$

with $[R_t]_{i,j} = 0$ for any $i, j \leq \rho$, and

$$[R_t]_{i,j} = \int_{x_{t-1}}^{x_t} \left[ p_t^{(\rho)} \right]_i \left[ p_t^{(\rho)} \right]_j dx \tag{33a}$$

$$\underset{(26)}{=} \int_{x_{t-1}}^{x_t} (x - x_{t-1})^{i+j-2\rho-2} \prod_{k=1}^{\rho} (i-k)(j-k) \tag{33b}$$

$$= \frac{(x_t - x_{t-1})^{i+j-2\rho-1}}{i+j-2\rho-1} \prod_{k=1}^{\rho} (i-k)(j-k), \tag{33c}$$

otherwise.

On the other hand, notice that by making use of the relations in (31) and (32), and given $e_0$, i.e., the vector that contains the initial $2\rho - 1$ spline derivative values at time stamp $x_0$ (for which the last $\rho - 1$ elements, i.e., $[e_0]_{\rho+1:2\rho}$, are zero-valued due to the natural boundary conditions of the solution stated in Section 2.1), the problem (25) becomes

$$\min_{\{a_t \in \mathbb{R}^{2\rho}\}_{t=1}^T} \left\{ \sum_{t=1}^T \ell \left( a_t^\top p_t(x_t), y_t \right) + \eta \, a_t^\top R_t a_t \right\} \tag{34a}$$

$$\text{subject to: } [a_t]_{1:2\rho-1} = e_{t-1}, \tag{34b}$$

$$[e_t]_i = \frac{1}{(i-1)!} \sum_{j=1}^{2\rho} [a_t]_j (x_t - x_{t-1})^{j-1} \prod_{l=1}^{i-1} (j-l). \tag{34c}$$

In practice, the nonzero-valued boundary conditions of the solution $[e_0]_{1:\rho}$ (or equivalently, $[a_1]_{1:\rho}$) are also learned. However, for the convenience of future derivations, we assume that their values are given within $e_0$.

Now, notice that every additive term in the cost (34a) along with their corresponding constraints in (34b) and (34c) can be constructed from the tuple $(x_{t-1}, x_t, y_t, e_{t-1}, a_t)$. This allows us to identify

an equivalent action-state problem representation for (34) where the combination coefficients $\boldsymbol{a}_t \in \mathbb{R}^{2\rho}$ are understood as actions and the $t$th signal state, defined as $\boldsymbol{\sigma}_t := (x_{t-1}, \boldsymbol{e}_{t-1})$, encodes the necessary information to resume the trajectory reconstruction from the previous time step $t - 1$, as illustrated in Figure 5. From this perspective the set of data points, i.e., $\mathcal{O}_{1:T} := \{\boldsymbol{o}_t\}_{t=1}^T$ with $\boldsymbol{o}_t := (x_t, y_t)$, is seen as a set of observations of the environment. Moreover, we can capture the signal state dynamics through

$$\boldsymbol{\sigma}_{t+1} = f_\sigma(\boldsymbol{\sigma}_t, \boldsymbol{a}_t; \boldsymbol{o}_t), \tag{35}$$

where $f_\sigma$ represents the signal state update mechanism. This is because $\boldsymbol{\sigma}_{t+1}$ is constructed from $x_t$ and $\boldsymbol{e}_t$ by definition, which in turn are obtained from the $t$th observation $\boldsymbol{o}_t$, and the $x_{t-1}$ time stamp (within $\boldsymbol{\sigma}_t$) and $t$th action $\boldsymbol{a}_t$ through (34c), respectively.

From here, problem (34) can be reformulated from an action-state perspective as

$$J(\boldsymbol{\sigma}_1; \mathcal{O}_{1:T}) := \min_{\{\boldsymbol{a}_t \in \mathcal{A}(\boldsymbol{\sigma}_t)\}_{t=1}^T} \left\{ \sum_{t=1}^T k(\boldsymbol{\sigma}_t, \boldsymbol{a}_t; \boldsymbol{o}_t) \right\} \tag{36a}$$

$$\text{subject to: } \boldsymbol{\sigma}_{t+1} = f_\sigma(\boldsymbol{\sigma}_t, \boldsymbol{a}_t; \boldsymbol{o}_t), \tag{36b}$$

where

$$\mathcal{A}(\boldsymbol{\sigma}_t) = \{\boldsymbol{a} \in \mathbb{R}^{2\rho} : [\boldsymbol{a}]_{1:2\rho-1} = \boldsymbol{e}_{t-1}\}, \tag{37}$$

denotes the set of actions (the spline combination coefficients) at time step $t$ that satisfy the $C^{2\rho-1}$ smoothness of the solution, and

$$k(\boldsymbol{\sigma}_t, \boldsymbol{a}_t; \boldsymbol{o}_t) := \ell\left(\boldsymbol{a}_t^\top \boldsymbol{p}_t(x_t), y_t\right) + \eta \, \boldsymbol{a}_t^\top \boldsymbol{R}_t \boldsymbol{a}_t, \tag{38}$$

is shorthand notation for each of the $t$th additive terms in the cost (34a). Moreover, viewing the minimal cost incurred in problem (36) as a mapping $J$ from signal states and sequences of observations to the real line is handly for applying the dynamic programming algorithm as we will show next.

First, let $\{\boldsymbol{a}_1^*, \ldots, \boldsymbol{a}_T^*\}$ be an optimal sequence of actions, which together with $\boldsymbol{\sigma}_1$ determines the corresponding optimal sequence of signal states $\{\boldsymbol{\sigma}_2^*, \ldots, \boldsymbol{\sigma}_T^*\}$ via the signal state dynamics (35). Now, consider the subproblem whereby we start at $\boldsymbol{\sigma}_t^*$ at time step $t$ and wish to minimize the cost-to-go from time step $t$ to $T$; that is, we would like to evaluate the optimal cost $J(\boldsymbol{\sigma}_t^*, \mathcal{O}_{t:T})$. Then, the truncated optimal action sequence $\{\boldsymbol{a}_t^*, \ldots, \boldsymbol{a}_T^*\}$ is optimal for this subproblem. This result is known as the principle of optimality (Bertsekas, 2019) and is the key concept that allows us to construct the optimal sequence of actions by evaluating (36) recursively as

$$J(\boldsymbol{\sigma}_t^*, \mathcal{O}_{t:T}) = \min_{\boldsymbol{a} \in \mathcal{A}(\boldsymbol{\sigma}_t^*)} \{k(\boldsymbol{\sigma}_t^*, \boldsymbol{a}; \boldsymbol{o}_t) + J(f_\sigma(\boldsymbol{\sigma}_t^*, \boldsymbol{a}; \boldsymbol{o}_t); \mathcal{O}_{t+1:T})\}. \tag{39}$$

## C  ADDITIONAL EXPERIMENTS

Here, we present some additional complementary experiments to Section 4.3 aiming at better enclosing the limitations while highlighting the strengths of our approach.

### C.1  SOME EXPERIMENT VARIATIONS

Figure 6 shows how a non-uniform sampling can affect the stability of the proposed strategy. Particularly, these graphs have been obtained under the same experimental setting as in Section 4.3 but with a percentage of $10\%$ missing data. Every missing data point creates a gap in time resulting in a non-uniform distribution of time stamps. As we can see, even the $H = 1$ data point forecasting strategies achieve stability.

Splines of a higher degree than the cubic spline, show unstable behavior even for the $L = 1$ lookahead (or Delay) policy, as illustrated in Figure 7. Luckily, this can be mitigated with a (slightly) larger prediction horizon $H$.

Finally, Figure 8 displays some of the sequentially interpolated trajectories discussed in the paper for visual intuition.

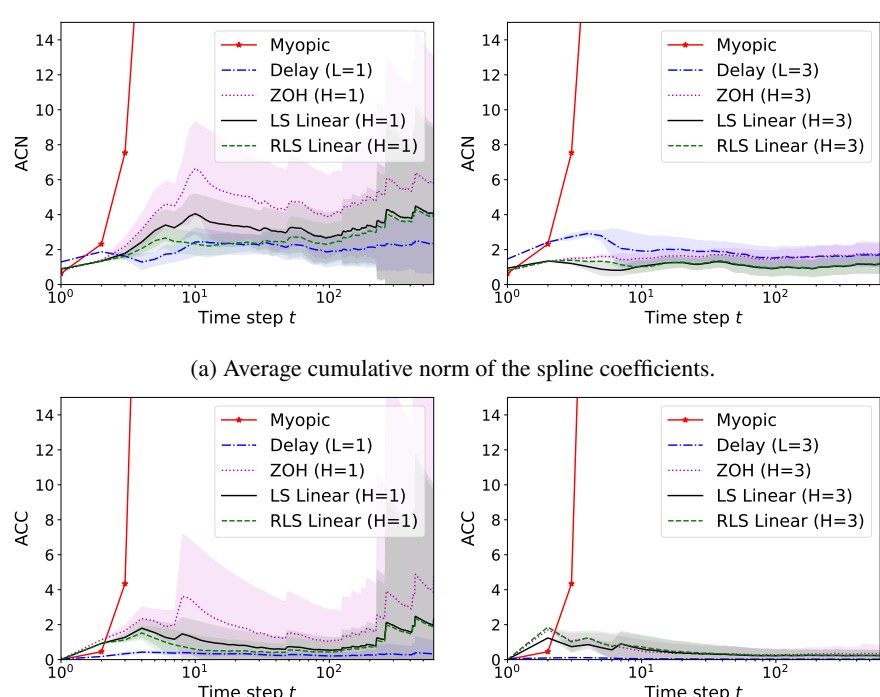

(a) Average cumulative norm of the spline coefficients.

(b) Average cumulative cost incurred by the interpolator.

Figure 6: Same configuration as in Figure 4 but over a sequence of non-uniformly arranged data points.

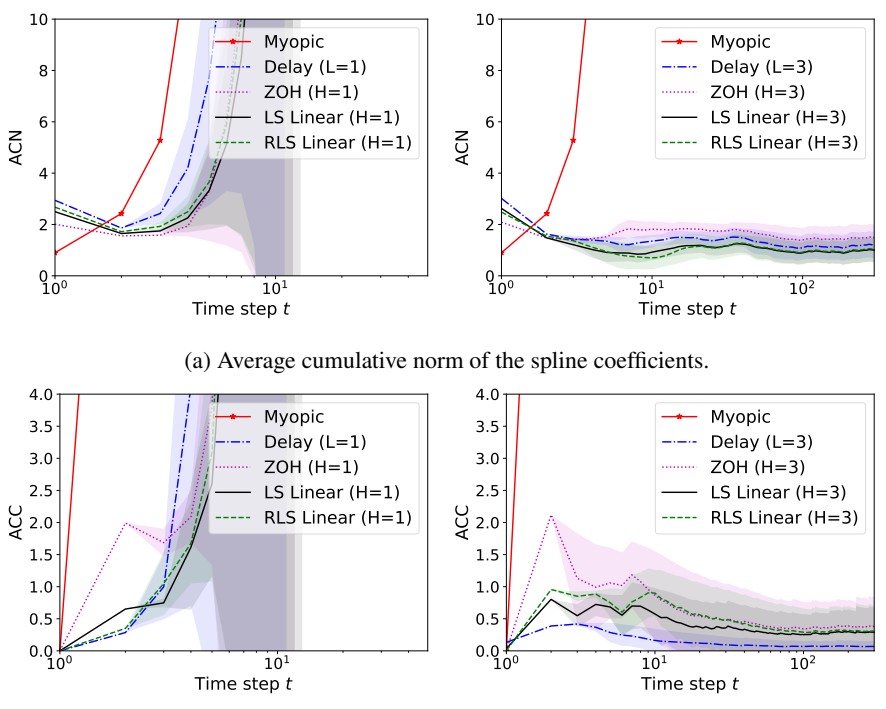

(a) Average cumulative norm of the spline coefficients.

(b) Average cumulative cost incurred by the interpolator.

Figure 7: Same configuration as in Figure 4 except for $\rho = 3$.

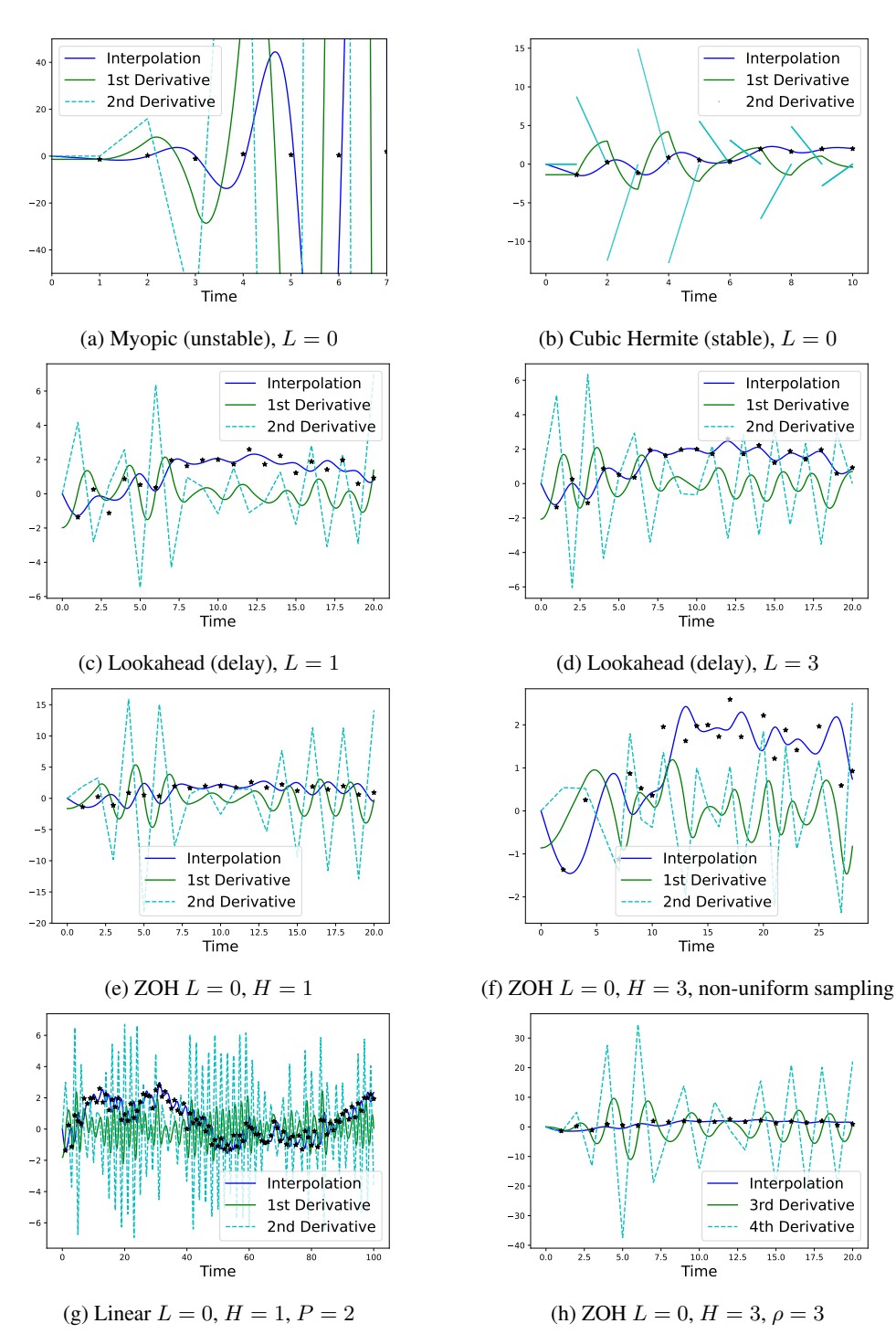

Figure 8: Visualization of sequentially interpolated trajectories for different configurations discussed along the paper. All of them for $\eta = 0.01$ and $\rho = 2$ except stated otherwise.

Table 1: Average execution time per interpolated piece for the experiment presented in Section 4.3. For context, this experiment has been run in an M3 Pro in Python.

| Interpolator | Execution time per interpolated piece (ms) | |
|---|---|---|
| Myopic | $\sim 3.4$ | |
| Delay | $\sim 5.5$ ($L = 1$) | $\sim 10.7$ ($L = 3$) |
| ZOH | $\sim 5.4$ ($H = 1$) | $\sim 10.4$ ($H = 3$) |
| LS Linear | $\sim 5.3$ ($H = 1$) | $\sim 10.5$ ($H = 3$) |
| RLS Linear | $\sim 5.5$ ($H = 1$) | $\sim 11$ ($H = 3$) |

Table 2: Total cost per interpolated piece for the experiment presented in Section 4.3.

| Interpolator | Execution time per interpolated piece (ms) | |
|---|---|---|
| Batch | 0.02495**797** | |
| Delay | 0.148 ($L = 1$) | 0.02495**873** ($L = 3$) |
| ZOH | 0.913 ($H = 1$) | 0.325 ($H = 3$) |
| LS Linear | 0.477 ($H = 1$) | 0.225 ($H = 3$) |
| RLS Linear | 0.448 ($H = 1$) | 0.240 ($H = 3$) |

### C.2 EXECUTION TIME PER INTERPOLATED PIECE

We have computed the execution time of each interpolator presented in Section 4.3. To do so, we have decorated the interpolation function with a decorator that measures the execution time (Ramalho, 2015). Table 1 shows the (average) execution time required to interpolate a spline piece. That is, it shows the prediction, reconstruction, and parameter training/parameter update time (if applicable) per time step.

As you can see in Table 1, all presented interpolator execution times per time step are roughly in the same order of magnitude.

> Notice that these results favor our proposed strategy since, for an affordable increase in execution time, e.g., compare Myopic (traditional approach unstable in this setting) vs. ZOH (ours), we can attain a certain degree of smoothness stably.

### C.3 SOME INSIGHTS INTO THE IMPORTANCE OF THE PREDICTOR'S ACCURACY AND HORIZON LENGTH

In Table 2, you can see the total cost per interpolated piece incurred by the interpolators in the experiment in Section 4.3.

For context, the total cost per interpolated piece equals the average cumulative cost (ACC) evaluated at the last time instant $T = 300$. For the sake of comparison, we have also added the total cost incurred by the optimal interpolator, i.e., the solution to (1), which cannot be computed before the whole batch of time-series data is available. For this reason, in Table 2 we refer to it as the batch interpolator.

As you can see in Table 2 the ZOH with a prediction horizon $H = 1$ performs the worst. This result is expected since it is arguably the simplest presented predictor that shows stability. From here, we have observed that the benefit of using a more accurate predictor, e.g., LS Linear, is not attaining stability more robustly but a reduction in the total incurred cost. Additionally, notice that any performance for a prediction horizon $H = 1$ is lower bounded by the interpolation with a delay $L = 1$ because it uses the actual data point to come (this is equivalent to perfect forecasting done by a hypothetical perfect predictor). Thus, the best case predictor would improve the ZOH strategy (the simplest) a $\sim 86\%$ instead of a $100\%$. This percentage is obtained by using the batch result

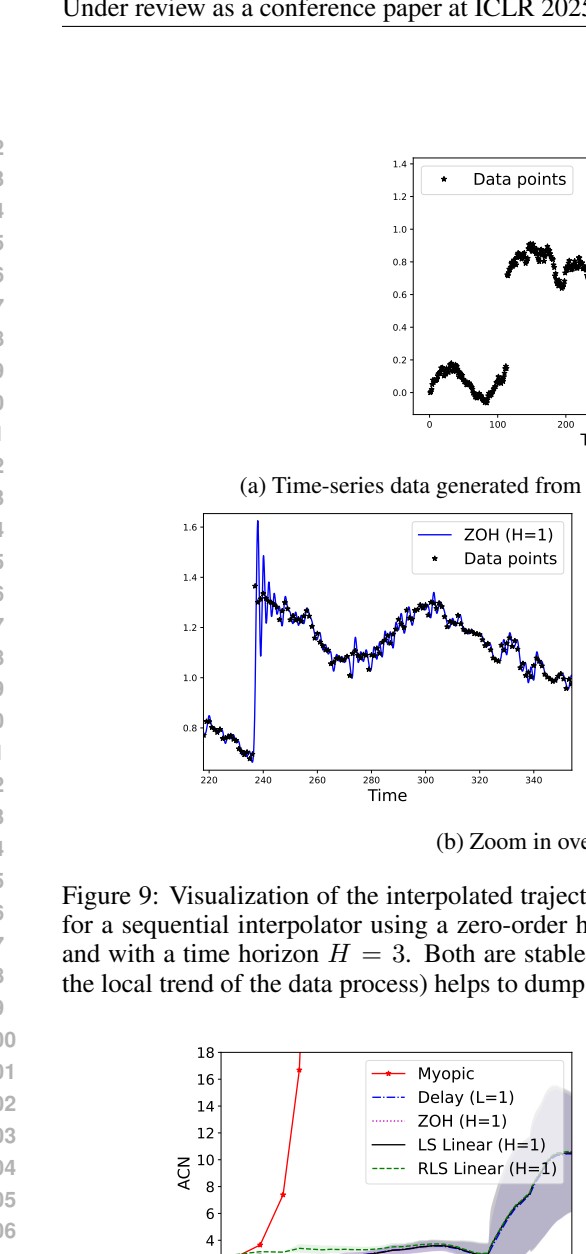

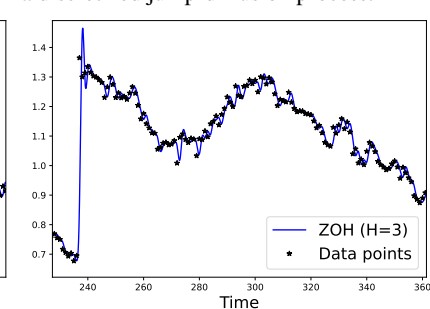

(a) Time-series data generated from a discretized jump-diffusion process.

(b) Zoom in over a jump region.

Figure 9: Visualization of the interpolated trajectory over the same time-series data, see Figure 9a, for a sequential interpolator using a zero-order hold (ZOH) predictor with a time horizon $H = 1$ and with a time horizon $H = 3$. Both are stable, but a higher prediction horizon $H$ (accentuating the local trend of the data process) helps to dump oscillations due to abrupt changes.

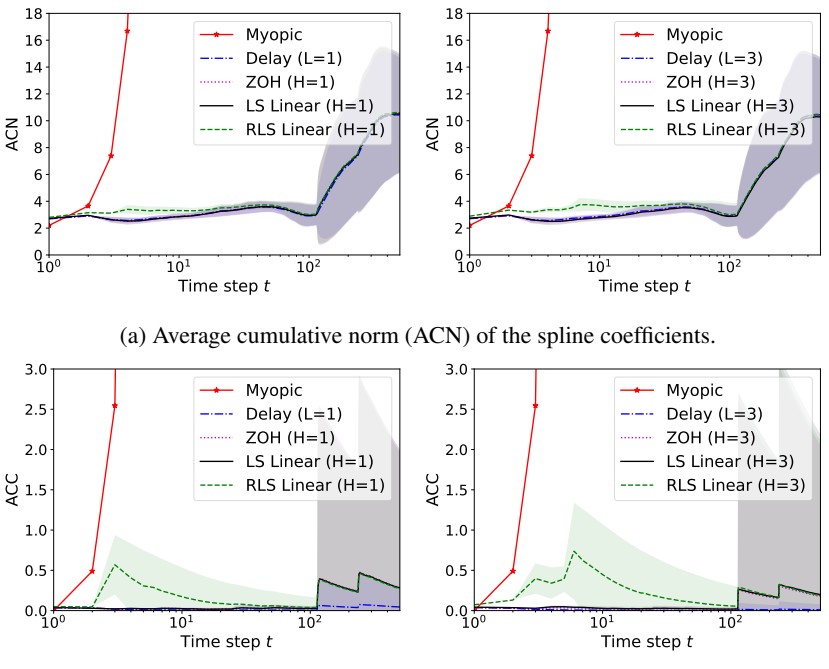

(a) Average cumulative norm (ACN) of the spline coefficients.

(b) Average cumulative cost (ACC) incurred by the interpolator.

Figure 10: ACN of the spline coefficients and ACC incurred by each interpolator for time-series data generated from a (discretized) jump-diffusion model. As expected from a stable behavior, both metrics are bounded in mean and average (except for the Myopic interpolator which is unstable).

(optimal) as a baseline, i.e., $(0.913 - 0.148)/(0.913 - 0.02495797) \simeq 0.86$. Of course, this best case can be improved by increasing the prediction horizon.

> Nevertheless, we have observed that usually a delay $L = 3$ is enough to perform close to optimally (in this case, a delay $L = 3$ matches the optimal result up to the 6th decimal value), and hence, the benefits of pursuing a perfect predictor for a larger horizon would be marginal.

### C.4 VALIDATION OVER A MORE CHALLENGING ENVIRONMENT

In order to validate the proposed interpolation strategy against a more challenging environment we have generated data from a (discretized) jump-diffusion process (Merton, 1976), i.e., a mixture of a diffusion process with a jump process. This time-series data may suffer sudden changes (or jumps) at random instants, e.g., see Figure 9a, which may pose a big challenge for a sequential spline interpolator under no delay and smooth constraints.

With this data, we have carried out similar (stability) experiments to the ones presented in Section 4.3, and the results are summarized in Figure 10. They corroborate the intuitive idea presented in the main document that capturing a trend of the streaming data can attain a stable reconstruction. That is, predicting a trend (or behavior) of the process has a similar (importance) weight in the stability than accurately predicting data fluctuations but with the advantage that is usually a simpler procedure.

> Note that this observation supports the use of simpler local prediction models in most environments. Especially, since most data processes can be accurately modeled locally by a constant or linear model.

Lastly, we would like to emphasize that the robustness of our strategy is possible due to the ability of spline models to adapt and approximate to local trends like constants or lines (regardless of the process generating the data). In alignment with this idea, and as an additional argument, the cubic Hermite splines can be proven to be internally stable (by the same procedure presented in Section 3.1) and always end an interpolated piece with linear behavior, i.e., the second derivative is zero at the end of the interpolated piece. See Figure 8b as an example. However, and differently from our strategy, this is done at the expense of sacrificing smoothness (it is only continuous up to the first derivative) in the reconstructed trajectory.

