# OpenReview forum: "Towards Stabilizable Sequential Smoothing Spline Interpolation by Point Forecasting"
_ICLR.cc/2025/Conference — ICLR 2025 Conference Withdrawn Submission_

### Official Review · Reviewer_eViq · 2024-11-02

**Soundness:** 2
**Presentation:** 2
**Contribution:** 2
**Rating:** 5
**Confidence:** 3

**Summary:**

I thank the authors for contributing this study to the community. In this paper, the author(s) proposed a new way of learning smoothing splines for a streaming dataset. The naive approach for this simply updates the smoothing spline using the just-observed data, and it is known to be potentially instable. The newly proposed approach predicts future observations using a parametric model trained on the past data, and then uses the available data and predicted data together to update the smoothing spline for the next time stamp.

The paper proved theories about stability of the underlying dynamics of spline coefficients and controllability of this dynamics. Besides the theory, a numerical test on a synthetic data is provided to justify that the proposed approach is more stable compared to the naive approach.

**Strengths:**

This paper studies an important question of learning a model from streaming data. This topic becomes more and more important in this big-data era. For the specific model considered in this paper -- the smoothing spline for streaming data -- the author(s) proposed a novel way of estimation, which is empirically stable and does not require waiting for future data like previous methods. This is demonstrated through an experiment in the paper. The authors also formulated the stability of the fitted spline as a dynamical system problem and provided some analysis on this aspect. This provides a novel approach to deepen the understanding of the instability issue.

**Weaknesses:**

The paper has a few technical and presentation issues. I will list the major ones below. I begin with technical issues. Below is a list of technical problems I think the author(s) should address in order for the paper to be valid/more readable.

1. In the proof of theorem 1, the authors seem to confuse the operator norm and spectral radius of matrices. It is not sufficient to conclude $\\|Ax\\| \leq c \\|x\\|$ from that the eigenvalues of A are smaller than c, even if A is positive upper-triangular. For example

\\begin{equation}A = \\begin{pmatrix} 1 & M\\\\ 0 & 1\\end{pmatrix}\\end{equation}

with some $M \gg 1$. Now take $x = (0, 1)^\top$ then you should observe $\\|A^k x\\|$ grows to infinity (in this case linearly in $k$).

2. The dynamics of the spline update follows equation (9), which is not the internally stable case studied in Theorem 1. The authors did not provide the connection between these two. What does internally stable/instable mean for the spline update? It is clear that internally instable implies (9) is instable if one can freely choose $\alpha_t^* = 0$, but I am not sure this is true from (10a) and (10b). Conversely, for $\alpha_t^*$ given as in (10a-b), can we say the "external" dynamics is necessarily instable? These important connections should be discussed if they are not trivial.

3. In section 3.2 the authors called $\alpha_t$ "inputs". I feel it is better if the authors can establish controllability using $o_t$ as inputs, showing that the spline segments can become any polynomial you want using appropriate data $(x_t, y_t)$. If $\alpha_t$ is used as inputs, then more explanation on what it means to "input" $\alpha_t$ to the system is helpful.

4. The methodology (19) is not clearly a valid choice. As the authors pointed out, the time series may not come in in a uniform time grid, and $u_t$ can change in $t$. Given this setup, I do not believe an AR-like model (19) can fit the data well. Suppose my data model is $y(t) = t$, and I observe $y_{t_1} = 1,~y_{t_2} = 2,\ldots, ~y_{t_k} = 2^{k-1}$ with time stamps $t_j = 2^{j-1}$. Then your learned $\\Theta$ would simply double the last observation by 2. But suppose now my $t_{k+1}$ is no longer $2^k$, then the prediction is completely wrong. So some justification of the model would be nice.

5. In view of 4 above, I don't think the numerical experiment is sufficient. First of all, the experiment uses an AR model to generate the data, for which one expects the method in (19) works fine. It is nice to see (19) works for some other models or for a real dataset. Second, for the AR data, I wonder if the windowing strategies mentioned in the paper can potentially work well, too. It would be nice if we can see comparisons between the new method and state-of-the-arts. Finally, the data is generated on a uniform time grid, and it is nice to see the performance of (19) on a non-uniform time grid.

Next I comment on some non-technical issues. The idea behind the work is nice, but the presentation can be better.

1. I understand these papers are quite compact, and some details cannot be elaborated. However, the main text should provide a smooth introduction to those not interested in the technical details and those new to the field. However, I found it occasionally hard to grasp the ideas looking at the main text, and I ended up looking at the appendix to confirm the "conditional expectation" of my interpretation of what the authors mean in the main text (e.g. the delay mechanism, the definitions in the first paragraph of section 2.2). To summarize, I don't think the authors did a perfect job presenting easy-to-understand overviews in the main text. To address this, I suggest the authors try to re-think the presentation in the main text, add in some formulas, and remove some text descriptions and pictures to save space (I like those, but in this case, they appear to be insufficient to convey the idea).

2. Some key parts of the paper is missing, making the logic of the paper less clear. (1) after you predict H steps into the future, how do your algorithm find the next segment? By doing a spline fit to the entire data? By solving the DP you constructed? How do you solve the DP if this is the case? A simple algorithm environment or even some description could be more concise than Figure 2. (2) How does the proposed approach improve stability? Theorem 1 says cubic splines are not internally stable, does your method alleviate this issue?

**Questions:**

Here are some other questions and comments, the major ones are in the Weakness section:

1. There are a few potential typos and off-by-one things in the text. For example, when defining matrix $A$ in equation (12), $A_{1,1}$ by the formula should be 0, not 1, and in (12) the matrix $A$ seems to be of shape $(2\rho+1) \times 2\rho$ following the ddots. The authors can carefully review them.

2. Maybe this is another formulation, but for the optimization problem (1), I usually see people using $D^{(\rho-1)}$ as the penalty term. Say for cubic splines, the penalty is the second derivative, like in (Hastie et al, The Elements of Statistical Learning, section 5.4).

3. It is not clear to me how do you solve the DP (5)-(8). Usually we propagate it backward in time, but seems (5)-(8) is forward in time. How do you evaluate J in (5) and optimize for $a_1^*$?

4. In section 4.2.1, how do you define $\mu_0$? Can you give some justification of your $\mu_t$?

5. Why is controllability important to know? What does it imply?

6. Does being stable necessarily imply not being controllable, and being controllable imply instable?

---

### Official Review · Reviewer_4KF1 · 2024-11-03

**Soundness:** 4
**Presentation:** 3
**Contribution:** 3
**Rating:** 8
**Confidence:** 4

**Summary:**

The paper addresses the instability in sequential smoothing spline interpolation, especially in low-delay scenarios where typical solutions sacrifice either delay or smoothness. Existing smoothing spline methods often depend on delaying data processing to stabilize trajectories, which is infeasible in real-time applications. This work introduces a novel stabilization approach through data forecasting, allowing low-delay operation without compromising smoothness. By formalizing the instability in sequential smoothing splines and establishing the controllability of these models, it fills the research gap in stabilizing real-time smoothing spline interpolation, especially in delay-sensitive contexts where both stability and smoothness are crucial.

The authors model the trajectory of a smoothing spline interpolator as a discrete dynamical system of spline coefficients, analyzing its internal instability and controllability. The primary strategy proposed for stabilization employs data point forecasting to predict future data points, simulating the effect of delayed data without waiting. The method leverages a dynamic programming approach to set up an action-update mechanism, with the instability and controllability of this mechanism analyzed through control theory. Two forecasting methods are explored: a simple zero-order hold model and a parametric linear model with optional online or offline learning. The strategy ensures stability by enabling forecasts that approximate the smooth behavior seen in delayed responses without actual delays, catering to low-delay regimes in sequential data​.

**Strengths:**

The treatment of the interpolation problem as a discrete dynamic estimation problem, allowing for application of forecasting, dynamic programming and application of formal stability and controllability metrics is a particular strength of this work.

The paper demonstrates originality by addressing a specific limitation in sequential smoothing spline interpolation, particularly under low-delay constraints where traditional methods fall short. A main novelty stems from the innovative use of data forecasting as a stabilization mechanism. Rather than requiring a delay or compromise on smoothness, as is common in existing methods, this approach leverages forecasting models to predict future data points, effectively simulating a delay without waiting. The paper rigorously evaluates the stabilization strategy in both uniformly and non-uniformly sampled data environments.

The authors provide theoretical foundation by formally proving the instability of sequential smoothing splines under low-delay conditions and presenting a forecasting-based stabilization strategy. The work is clearly written and significant in terms of potential applications.

**Weaknesses:**

1. While the work shares a valuable perspective, which - in the opinion of this reviewer - moves beyond previous attempts, certain weaknesses are identified.
Firstly, the original work where the interpolation problem is cast as one of dynamic programming is the one by Bellmann, Kashef and Vasudevan (1972), which is not cited or discussed in this work. The links to that original work and how this work moves forward need to be clarified.
Similarly, works on zero delay interpolation using alternate trainable strategies exist and some mention of this is warranted, e.g., Ruiz-Moreno, Lopez-Ramos,  Beferull-Lozano (2023).

2. While the paper introduces forecasting as a core strategy for stabilization, the exploration of forecasting models is somewhat limited. This is in part justified, since the main devised experiments are generated by linear AR(2) models, which allows for use of a basic zero-order hold model or a simple linear model. However, at the same time, this likely means that a training of an AR forecasting model could suffice for the task at hand. Given the simple case studies, the work not delve into exploration of more sophisticated forecasting techniques. It is of course appreciated that section C4 is added to tackle this consideration, however, it seems like this work would benefit from inclusion of more such complex processes.

3. The paper would benefit from more explicit comparisons to contemporary interpolation approaches that also aim for zero delay. While the method’s novelty is highlighted against traditional delay-based techniques, the work does not sufficiently benchmark its performance against other recent zero-delay interpolation methods, such as the trainable real-time interpolators (RTIs) discussed in the referenced work from 2022. Such comparisons could validate the claimed advantage of this approach in real-world applications.

4. Given the relevance of this work for interpolation of real-world datasets, it is surprising that no such datasets are actually employed. For instance, the aforementioned work of  Ruiz-Moreno, Lopez-Ramos, Beferull-Lozano (2023) employs one synthetic dataset and five
real datasets. It seems that such real datasets can also be considered herein.

**Questions:**

1.  It seems that the performance of the adopted forecasting method is critical to the performance of the proposed scheme. In fact, the such step-ahead estimation schemes are often described by state space forms, which come with own stability considerations. How is this accounted for?

2. In the same vein, could you discuss potential limitations associated with forecasting errors in your method? How do you handle scenarios where the forecasting model is inaccurate, and could forecast errors potentially destabilize the interpolation?

3. It seems that certain hyper-parameters are involved in the proposed setting. How are these configured. Is a sensitivity analysis necessary to ensure adequate performance?

4. Could you provide more details on the computational overhead introduced by forecasting at each step? Specifically, how does the computational load vary with forecasting model complexity, and what measures are taken to ensure real-time applicability?

---

### Official Review · Reviewer_r6eK · 2024-11-03

**Soundness:** 1
**Presentation:** 2
**Contribution:** 1
**Rating:** 3
**Confidence:** 3

**Summary:**

The submission discusses the stability of sequential implementations of spline smoothing.
The main contributions are:

- Identifying a time-varying linear system for the smoothing spline coefficients through dynamic programming
- Analysing the stability and controllability of this system to study the stability of the spline interpolation problem
- Stabilising spline interpolation through delay and forecasting

The experiments suggest that the stabilisation strategies improve the stability of spline interpolation notably.

**Strengths:**

1. The introduction is nicely written and easy to follow.
2. The experiments are convincing, especially those in Appendix C.
3. A paper that studies the interface of splines, dynamic programming, and state-space models should provide relevant insights to the broader machine learning community (even if the technical results are heavy on control theory and signal processing, which not every machine learner might be familiar with).

**Weaknesses:**

Unfortunately, I recommend rejecting the submission despite the strengths outlined above. The reason is that the stability analysis raises questions which I doubt can be resolved without major revisions.
Concretely, I identify the following weaknesses.

### 1. Result 1 seems incorrect

- Equation 16 in the proof of Result 1 needs to be explained more thoroughly. I need more instructions to verify that Equation 16 is the correct determinant. Further, the statement that det(M) can't be zero for any u_t > 0 is incorrect: take $u_2=u_3=1$ and $u_4=4$, then Equation 16 is zero.
- I don't think the system in Equation 12 is controllable. For example, take $\rho=2$ and $u_t=1$ for all $t$ so that time-invariant theory applies. Then, A has left-eigenvector $(0, 0, 1, 3)$ with eigenvalue 2, and this eigenvector has a nonzero inner product with $B$. Thus, the setup contradicts the condition in Appendix C.6.3 in Anderson and Moore (1979). It also contradicts conditions C.5.3 (reachability) and C.7.3 (stabilisability). By padding with zeros and replacing 1 and 3 appropriately, the same case can be made for $\rho > 2$.
- Conjecture 1 is claimed to be supported by simulation, but the simulation results are not in the paper.

### 2. The linear-system perspective needs more clarity

- The linear system in Equation 9 (with parameters in Equation 12) is a central contribution of the paper according to the "Contribution" paragraph on page 2. As a reviewer, I need more instructions for deriving Equation 9 from Equation 7 than in line 201. As is, I can't verify or falsify Equation 9, which is problematic because Result 1 builds on Equation 12, and Result 1 contains mistakes (see previous point).
- The terminologies of controllability, reachability, and stabilisability need to be more clearly distinguished. Section 3 uses "controllable", but Result 1 shows "reachability", and Section 4 interprets Result 1 as having shown stabilisability. For context, I use the terminology from Appendices C.5, C.6, and C.7 in Anderson and Moore's "Optimal filtering" book (1979).



### 3. The manuscript lacks a discussion of spline smoothing and linear systems via stochastic processes

Another connection between linear systems and smoothing splines is known (via stochastic processes, not via dynamic programming):

> Kohn, Robert, and Craig F. Ansley. "A new algorithm for spline smoothing based on smoothing a stochastic process." SIAM Journal on Scientific and Statistical Computing 8.1 (1987): 33-48.

> Wahba, Grace. "Improper priors, spline smoothing and the problem of guarding against model errors in regression." Journal of the Royal Statistical Society Series B: Statistical Methodology 40.3 (1978): 364-372.

Wahba relates the smoothing spline to the repeatedly-integrated Wiener process. Concrete expressions for the time-discretisations of integrated Wiener processes (compare these to Equation 12 in the submission) are in Section 5.4 of:

> Hennig, Philipp, Michael A. Osborne, and Mark Girolami. "Probabilistic numerics and uncertainty in computations." Proceedings of the Royal Society A: Mathematical, Physical and Engineering Sciences 471.2179 (2015): 20150142.

Both Wahba (1978) as well as Kohn and Ansley (1987) need to be discussed more prominently. I mention Hennig et al. (2015) because Hennig et al.'s Section 5.4 eases comparing Equation 12 in the main paper to Wahba's work.

**Questions:**

Here are some minor questions and comments. They do not affect my score because they are far less important than the points made above and because I think they should be easy to resolve. Still, I think resolving them might improve the manuscript.

- What's the setup for Figure 1?
- There are inconsistent linebreak formats (e.g. line 062/063 versus line 071/072). Is this on purpose?
- Line 077: What do "overlapping" and "distort" mean in this context?
- Line 081: What does "this representation choice" refer to? (This question might appear petty, but this sentence seems critical for the problem setting).
- Line 307/308: Where are the results for this statement?
- The problem is likely a lack of knowledge on my side, but I struggle to connect Section 2.2 ("Dynamic programming approach") to

    > Bellman, R., B. G. Kashef, and R. Vasudevan. "Splines via dynamic programming." Journal of Mathematical Analysis and Applications 38.2 (1972): 471-479.

    It might also be reasonable to expect a discussion of Bellman et al. in the paper, which I couldn't find. If the authors agree with me, it would be great to see it added to the paper.

- The parametrisations of A in line 216 versus Equation 12 do not match. Line 216 suggests that A has zeros on the diagonal and constant factors (independent of u) on the first upper diagonal.

---

### Official Review · Reviewer_QZE6 · 2024-11-04

**Soundness:** 3
**Presentation:** 3
**Contribution:** 3
**Rating:** 6
**Confidence:** 4

**Summary:**

The authors use dynamical systems theory to understand the problem of stabilizing smooth spline interpolations in low-delay situations. This work formalizes the internal instability and asserts the controllability of sequential smoothing spline interpolators. They provide a stabilizing strategy based on data point forecasting capable of operating even under delay-less regimes and without sacrificing any smoothness of the interpolated trajectory

**Strengths:**

1. This paper is a good first work in exploring the problem of stabilization for smooth spline interpolations through the lens of dynamical systems theory.
2. The effect of the delay of the sequence of data points on the action sequence (modeled as a limited lookahead) is clearly motivated and validated through the theory and experimental results.
3.  Overall, an interesting connection between two well-established fields of spline interpolation and dynamical systems theory with good applications in forecasting.
4. The sequential nature of the splines is a practical set-up for large-scale data throughput (i.e. memory constraints).

**Weaknesses:**

1. The important hyperparameters that are paramount to determining roughness and solution trade-off contribute to solution stability, so I am not sure why these are being fixed instead of learnt adaptively. This approach seems a bit limited in that these are not varying coefficient models.
2. The authors addressed this already, but their Conjecture in that the system is controllable for any \rho needs more theoretical foundation. This seems to be the crux but needs much more foundation.
3. More discussion could be had on the offline setting (where the parameters are actually being trained) but this is not that big a deal for the context of this paper.

**Questions:**

1. If these hyperparameters are assumed given for the scope of this paper, how did you set them? Since these tune the tradeoffs between roughness and fit quality, it seems like these are something very important to determine stability. Perhaps there is a way to adaptively update these parameters to determine the forecasts?
2. I understand the contribution is to use dynamical systems theory to analyze the instabilities in the low-latency regime, but how does this compare to adaptive (for the parameters) methods such as Bayesian p-splines?
3. The state dynamics are then Markovian in nature? That seems like it should be mentioned if it is, please correct me if I'm wrong in assuming.
4. Can you explain briefly why you are considering a linear dynamics model (i.e. since the actions a_t are given by a dynamical system featuring matrices that capture certain relationships). Is this sufficient for the model's expressive power, or are you simply leveraging the well-established linear control theory as a start?

---

### Author Response · Authors · 2024-11-16

We would like to thank the reviewers for their time and effort. Their comments, concerns, and positive feedback are truly valuable.

This review has been really helpful for us to reaffirm that the research problem discussed in this manuscript is of interest to the community. It has also made us realize that the current literature review needs to be expanded and the technical claims refined.

Overall, we believe that our findings are promising, and we would like to do them justice in the manuscript. For this reason, we have decided to withdraw the paper and work towards an improved version.

---

### Note · Authors · 2024-11-16

I have read and agree with the venue's withdrawal policy on behalf of myself and my co-authors.